# Optimization of Mouse Growth Hormone Plasmid DNA Electrotransfer into Tibialis Cranialis Muscle of “Little” Mice

**DOI:** 10.3390/molecules25215034

**Published:** 2020-10-30

**Authors:** Eliana Rosa Lima, Claudia Regina Cecchi, Eliza Higuti, Gustavo Protasio Pacheco de Jesus, Alissandra Moura Gomes, Enio Aparecido Zacarias, Paolo Bartolini, Cibele Nunes Peroni

**Affiliations:** 1Biotechnology Center, Instituto de Pesquisas Energéticas e Nucleares (IPEN-CNEN), Cidade Universitária, 05508-000 São Paulo, SP, Brazil; lima-eliana@hotmail.com (E.R.L.); crcecchi.ipen@gmail.com (C.R.C.); eliza_higuti@yahoo.com.br (E.H.); gpjesus88@gmail.com (G.P.P.d.J.); li.moura@hotmail.com (A.M.G.); Bio_Enio@hotmail.com (E.A.Z.); pbartoli@ipen.br (P.B.); 2Translational Neuropsychiatry Unit, Department of Clinical Medicine, Aarhus University, 8000 Aarhus, Denmark; 3Biotechnology Quality Control Laboratory, Butantan Institute, 05503-900 São Paulo, SP, Brazil

**Keywords:** electrotransfer, gene therapy, homologous model, mIGF-1, little mice, mouse growth hormone, non-viral gene transfer, tibialis cranialis muscle

## Abstract

Previous non-viral gene therapy was directed towards two animal models of dwarfism: Immunodeficient (lit/scid) and immunocompetent (lit/lit) dwarf mice. The former, based on hGH DNA administration into muscle, performed better, while the latter, a homologous model based on mGH DNA, was less efficient, though recommended as useful for pre-clinical assays. We have now improved the growth parameters aiming at a complete recovery of the lit/lit phenotype. Electrotransfer was based on three pulses of 375 V/cm of 25 ms each, after mGH-DNA administration into two sites of each non-exposed tibialis cranialis muscle. A 36-day bioassay, performed using 60-day old lit/lit mice, provided the highest GH circulatory levels we have ever obtained for GH non-viral gene therapy: 14.7 ± 3.7 ng mGH/mL. These levels, at the end of the experiment, were 8.5 ± 2.3 ng/mL, i.e., significantly higher than those of the positive control (4.5 ± 1.5 ng/mL). The catch-up growth reached 40.9% for body weight, 38.2% for body length and 82.6%–76.9% for femur length. The catch-up in terms of the mIGF-1 levels remained low, increasing from the previous value of 5.9% to the actual 8.5%. Although a complete phenotypic recovery was not obtained, it should be possible starting with much younger animals and/or increasing the number of injection sites.

## 1. Introduction

The non-viral gene delivery system using plasmid DNA that in most cases is injected directly into certain tissues, particularly muscle, produces significant levels of gene expression, although typically lower than those achieved with viral vectors. Nonetheless, this type of gene therapy is among the most frequently used in clinical trials, representing 16.6% of the worldwide protocols in 2017, close behind adenovirus (20.1%) and retrovirus (17.9%) administrations [1]. Non-viral methods have several advantages compared with viral vectors, including relative safety, the ability to transfer much larger genes, less toxicity and easier DNA preparation [2]. Its relatively low transfection efficiency and poor transgene expression can be substantially increased when plasmid DNA administration is followed by electroporation [1,2,3,4,5,6]. The first human trial of gene transfer using electroporation was reported in 2008 for an interleukin-12 plasmid delivered into the tumors of patients suffering metastatic melanoma [7]. Close to 90 clinical trials involving electrotransfer have been performed subsequently and the technique is considered to be relatively inexpensive, flexible and safe because it does not introduce any additional chemicals or viruses into cells [8]. However, as far as we know, the utilization of naked DNA is still not at the clinical level for systemic diseases [5].

To set up gene therapy protocols based on growth hormone plasmid DNA administration, we have utilized immunodeficient (lit/scid) or immunocompetent (lit/lit) dwarf mice in order to investigate electrotransfer of human (hGH) or mouse (mGH) genes into muscle [4,5,9,10,11]. In these studies, we routinely encounter better results with the administration of hGH DNA to immunodeficient dwarf mice. In previous work, the injection of the hGH gene into the tibialis cranialis (TA) muscle of 40 day old lit/scid mice provided a high catch-up growth of 76% for femur, 36% for nose-to-tail length and 39% for tail length [5]. Further studies on the different in vivo expressions of either hGH or mGH in these mice models could also take advantage of a recent review in which partially humanized transgenic mice were able to co-express hGH and mGH, offering a unique opportunity of investigation [12].

In the present study, we have carried out an initial optimization of this methodology to improve the transfection efficiency in the homologous model based on genomic mGH DNA electrotransfer into the tibialis muscle of immunocompetent lit/lit mice, taking into account the recommendation of the use of this model for setting up preclinical testing of non-viral gene therapy.

## 2. Results

The High Voltage/Low Voltage (HV/LV) electrotransfer protocol, based on one 800 V/cm pulse of 100 μs followed by one 400 ms pulse of 100 V/cm [11,13], was compared to a new protocol based on three 375 V/cm pulses of 25 ms each [14,15], providing the results shown in Figure 1A, where the advantage of injecting a volume of 20 μL was also confirmed. The higher efficiency obtained with the 375 V/cm protocol led us to adopt this strategy for the subsequent experiments. Moreover, because we had demonstrated the advantage of transfecting the tibialis cranialis muscle rather than the exposed quadriceps [11] this modification was also maintained.

Although 50 μg of DNA was routinely used in each injection [4,5,9,10,11], it was necessary to confirm whether this was still the optimum amount of DNA to be administrated under the new conditions. As shown in Figure 1B, the new conditions had no effect on the optimum amount of DNA and a slightly higher transfection efficiency, though statistically non-significant, was still obtained with 50 μg DNA per injection.

In contrast to the lack of an improvement upon increasing the amount of DNA per injection, a substantial increase in expression efficiency was obtained upon increasing the number of injection sites. This result is presented in Figure 2A, where 14.7 ± 3.7 ng/mL vs. 0.5 ± 0.2 ng/mL for saline (*n* = 3 for each group; *p* < 0.001) were obtained when the approach of two injections into each tibialis cranialis muscle was employed, in comparison with two injections into a single tibialis cranialis muscle (5.9 ± 2.7 ng/mL, *p* < 0.02, *n* = 3). Even higher levels of up to 21.1 ± 1.0 ng hGH/mL vs. 1.1 ± 0.1 ng/mL for the control group (*n* = 7 for each group; *p* < 0.001) were obtained in the serum of lit/scid mice three days after two injections of genomic hGH DNA plasmid into each tibialis cranialis muscle (Figure 2B).

Based on these results a 36-day bioassay was carried out utilizing 60-day-old mice: 5 lit/lit mice as a saline-injected control; 7 lit/lit mice as the DNA-treated group and 5 lit/+ mice of the same strain and age as a positive control. Figure 3 shows the data for the variation of the weight of each group during the bioassay and the figure legend summarizes the best fit second-degree or quadratic polynomial fits of these data. The differences in the slopes of the three curves, as measured by the linear coefficient (B1) of each equation, are particularly evident and reflects the fact that the weight gain for the treated mice is significantly different (*p* < 0.0001) from that of the untreated negative control. The difference in slope between the curves for treated lit/lit and for heterozygous co-aged mice was non-significant.

In addition to secretion efficiency, there was also a good sustainability of the mGH serum levels at the end of the 36-day assay. In DNA-treated mice, the average value of 8.5 ± 2.3 ng/mL (*n* = 7) was significantly higher (*p* < 0.02) than that presented by heterozygous mice (4.5 ± 1.5 ng/mL, *n* = 4) (Figure 4).

Table 1 summarizes all of the important parameters obtained during the 36-day bioassay. Particularly noteworthy are the results for the catch-up growth, which range from 38 to 83%. These values are much higher than those obtained in previous work on lit/lit mice (14 to 24%) [10] and are comparable to the range of 36–76% for the catch-up growth obtained for lit/scid mice in a 60-day assay with 40-day old mice subjected to two injections per animal (on day 1 and on day 41) of hGH plasmid DNA [5].

In Table 2 the mIGF-1 serum levels determined in lit/lit and heterozygous mice at the end of the 36-day bioassay are presented. In the present case, with 60-day old lit/lit mice, the final mIGF-1 catch-up was better than the 5.9% obtained in previous work with 60-day old lit/lit mice [10], but still only a relatively poor 8.5%.

## 3. Discussion

The newly set up electroporation and assay conditions provided, as far as we know, the highest circulatory levels of hGH in lit/scid and of mGH in lit/lit ever reported for GH non-viral gene therapy and will greatly facilitate the utilization of the homologous model for pre-clinical assay.

Concerning the initial electrotransfer protocol (Figure 1A), it was considered to be more practical and more efficient to carry out this first optimization in a 3-day assay, using a single hGH DNA plasmid administration to lit/scid mice. In previous studies, this strategy always provided relatively high values of the hormone expression levels [4,5,9,11] when compared to the homologous model based on mGH DNA electrotransfer [10].

Circulatory levels of hGH (21.1 ng/mL) and of mGH (14.7 ng/mL) confirm the high efficiency of the technique adopted, which, as previously observed, generally gives better results for hGH DNA administration in immunodeficient dwarf mice. A higher expression efficiency of a multi-location injection has also been observed by other authors for leptin gene electroporation [16].

The efficient mGH DNA transfection protocol was tested, moreover, via a 36-day bioassay, obtaining a weight variation curve that was significantly different from that of the untreated negative control and quite similar to that for the natural weight increase of co-aged heterozygous mice. The absolute daily weight increase of the treated animals appears in fact to be slightly above that of normal mice, though non-significantly different, suggesting that treated lit/lit and co-aged heterozygous underwent comparable growth in terms of absolute body mass. A similar situation was previously observed for 80 day old lit/scid treated in an analogous fashion [5]. In the present case, the lit/lit mice, with 60 days of age at the start of the bioassay, had a body weight 1.6-fold lower than the normal co-aged mice and, even more important, had practically passed the pubertal growth spurt determined by GH spikes and the consequent natural rise in serum IGF-1 [10,17,18]. Thus, although gene therapy-induced body weight gain and body length recovery clearly occurred, by 60 days after birth, the therapy could no longer cause a complete catch-up or restore the correct phenotype.

It should also be emphasized that, after 36 days, DNA-treated mice still presented serum levels of 8.5 ng mGH/mL, while in previous long-term bioassays these values were always in the range of 1.5–4.6 ng/mL for either hGH or mGH plasmid DNA administration [4,5,10].

As indicated by the data in Table 2, low mIGF-1 serum levels, which are strictly related to mice age and puberty [5,19], continue to be a potential limitation not yet fully resolved by our treatment. In the previous study with 40-day old lit/scid mice [5], the mice exhibited 250 ng/mL of mIGF-1 when assayed on day 15 against 277 ng/mL mIGF-1 for co-aged normal mice, indicating a “temporarily almost complete catch-up”! Unfortunately, these levels remained at 215 ng/mL vs. 403 ng/mL at the end of the 60-day bioassay, reducing catch-up from 88% at day 15 to only 48% at day 60.

## 4. Materials and Methods

### 4.1. Animals

The mutant strains of CB17-Ghrh lit/+ Prkdc scid/Bm (lit/scid) and C57BL/6J-GHRHR^LIT/+^ (lit/lit) mice were obtained from the Jackson Laboratory (Bar Harbor, ME, USA) [20]. The animals were maintained on a vented shelf and used to breed colonies of lit/scid and lit/lit mice, as previously described [21,22]. Males and females of approximately 60 days of age were utilized for the experiments, which were approved by the local ethics committee. Heterozygous lit/+ mice were used as positive control animals.

### 4.2. Plasmids

The plasmid pUC-UBI-hGH, containing the human Ubiquitin C promoter and the genomic hGH sequence of 2.1 Kb, has already been described in previous works [4,5,21,23]. The plasmid pUC-UBI-mGH, which encodes the genomic mGH sequence of 1.6 Kb, was obtained by substituting hGH-gDNA with mGH-gDNA [10]. These plasmids were multiplied in DH5α bacteria and purified using the *Nucleobond Xtra-midi* kit from Macherey-Nagel (Düren, Germany).

### 4.3. Plasmids Administration and Electroporation

Lit/scid or lit/lit mice were anesthetized with a mixture of xylazine and ketamine, followed by hyaluronidase (20 U/20 µL) injection into the non-exposed tibialis cranialis muscle. After 30 min, plasmid DNA (50, 75 or 100 µg) in different volumes (10 or 20 µL) was administered in the same region, followed by electrotransfer. Two protocols were used for electroporation: One 800 V/cm pulse of 100 µs followed by one 100 V/cm pulse of 400 ms, called “High Voltage/Low Voltage” (HV/LV) [16] and already used in our laboratory [5,10], or the new protocol adopted in the present work based on three 375 V/cm pulses of 25 ms each [14,15]. An ECM-830 electroporator and a caliper electrode with 3 mm distance between the plates (length/size 1.0 × 1.0 cm), both from BTX (Holliston, MA, USA), were utilized. Injection of saline was used as the negative control in all assays. A comparison between two injections into one TA and two injections into each TA muscle was also carried out. In short term assays, blood was withdrawn for determination of serum hGH or mGH after three days. For the analysis of in vivo hGH expression, an in-house radioimmunoassay, based on NIDDK reagents (Dr. A.F. Parlow, National Hormone and Pituitary Program, Torrance, CA, USA), whose sensitivity is 0.1 ng/mL, was used [22]. Mouse GH concentrations were determined by the *Rat/Mouse Growth Hormone ELISA* kit from Millipore (St. Charles, MO, USA).

### 4.4. Bioassay

A 36-day bioassay was carried out using two injections of the mGH-DNA plasmid into each tibialis cranialis muscle, followed by electroporation via the three 375 V/cm pulses of 25 ms protocol. Three groups of mice were used: one group of lit/lit mice as the saline-injected control (*n* = 5); a second group of lit/lit mice as the DNA-treated group (*n* = 7), and a third group of heterozygous (lit/+) mice as a positive control (*n* = 4). The body weight of the animals was determined throughout the entire assay period and used to calculate the average daily weight variation. The nose-to-tail length was measured with an electronic caliper before and at the end of the assay, when blood was collected to determine mGH and mIGF-1 concentrations. Serum mIGF-1 levels were measured using the *Quantikine mouse-rat IGF-1* kit (R&D Systems, Minneapolis, MN, USA).

Right and left femur lengths were measured before and at the end of the experiment by radiography.

### 4.5. Radiographic Measurements

An in vivo Imaging System FXPRO from Bruker (Billerica, MA, USA), available at the Facilities Center for Research (CEFAP, Institute of Biomedical Sciences, University of São Paulo, São Paulo, SP, Brazil), was used for measuring right and left femur length, before and at the end of the bioassay, as described [11]. Mice were anesthetized with a mixture of xylazine and ketamine for image acquisition. Images were analyzed via the Image J software.

### 4.6. Catch-Up Growth Calculation

Catch-up growth (C-uG) was calculated, as previously reported [5,9], using the body weight (g), nose-to-tail (cm), femur (mm) lengths or mIGF-1 (ng/mL) concentration, according to the following Formula (1):

C-uG = (Wt − Wc)/(Wn − Wc) × 100
(1)
where: Wt = final weight or length or mIGF-1 concentration of the treated group; Wc = final weight or length or mIGF-1 concentration of the control (saline-treated) group; Wn = final weight or length or mIGF-1 concentration of a normal co-aged animal group (heterozigous lit/+ mice, in this case).

### 4.7. Statistical Analyses

Quantitative variables, reported as the mean ± SD, were analyzed via the unpaired Student’s t test. Growth equations were generated by fitting the data to a quadratic relationship. The quadratic, linear and independent coefficients, calculated for each of the different experimental groups, were compared via the F-test method included in the Prism 5.0 package (GraphPad Software Inc., La Jolla, CA, USA). Means and best-fit curves were considered to be statistically different, representing distinct treatment effects, when the *p* value was < 0.05.

## 5. Conclusions

In this work, the conditions for the treatment of growth hormone deficient lit/lit mice have been greatly improved. The highest GH circulatory levels ever reported for a non-viral gene therapy protocol were attained. A more precocious treatment, starting with pre-pubertal mice and possibly including additional injection sites, has the potential, in our opinion, of providing a complete phenotypic recovery. Future bioassays with <40-day old little mice are currently being planned, noting that this gene therapy treatment may be particularly delicate.

## Figures and Tables

**Figure 1 molecules-25-05034-f001:**
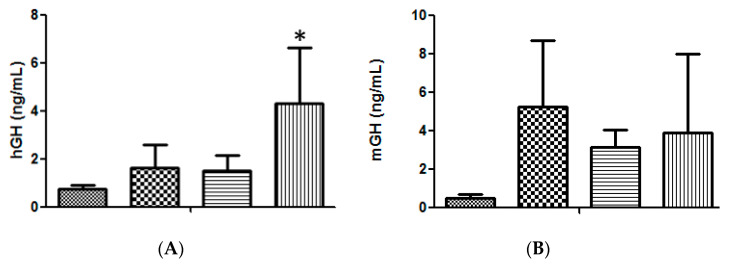
(**A**). Serum hGH concentration in lit/scid mice, 3 days after a single administration of 50 μg of hGH plasmid DNA, in different volumes and under different electrotransfer conditions, into the right tibialis cranialis (TA) muscle: (
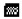
) negative control (20 μL saline); (
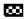
) 50 μg DNA in 10 μL applying the 375 V/cm protocol; (
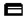
) 50 μg DNA in 20 μL applying the HV/LV protocol; (
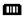
) 50 μg DNA in 20 μL, applying the 375 V/cm protocol. (*n* = 6 animals per condition); significance test, in comparison with the negative control: * *p* < 0.02. (**B**). Serum mGH concentration in lit/lit mice, 3 days after a single administration into the right TA muscle of different amounts of mGH plasmid DNA dissolved in 20 μL of saline, followed by the 375 V/cm electrotransference protocol: (
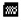
) negative control (saline); (
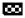
) 50 μg DNA; (
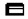
) 75 μg DNA; (
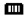
) 100 μg DNA. (*n* = 3 animals per condition).

**Figure 2 molecules-25-05034-f002:**
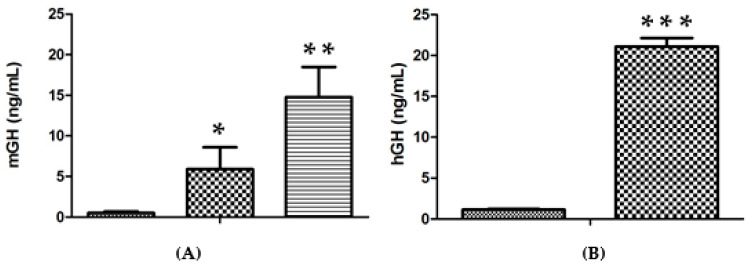
(**A**). Serum mGH concentration in lit/lit mice, 3 days after administration of 50 μg mGH plasmid DNA per injection (20 μL) in the TA muscle, followed by electrotransfer based on the 375 V/cm protocol: (
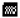
) Negative control (saline); (
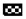
) two injections into one TA muscle; (
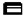
) 2 injections into each TA muscle (*n* = 3 animals per condition). (**B**). Serum hGH concentrations in lit/scid mice, 3 days after administration of 50 μg hGH plasmid DNA per injection (20 μL) in the TA muscle followed by electrotransfer based on the 375 V/cm protocol: (
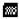
) negative control (saline); (
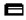
) 2 injections into each TA muscle (*n* = 7 animals per condition). Significance test, in comparison with the negative control: * *p* < 0.05; ** *p* < 0.005; *** *p* < 0.001.

**Figure 3 molecules-25-05034-f003:**
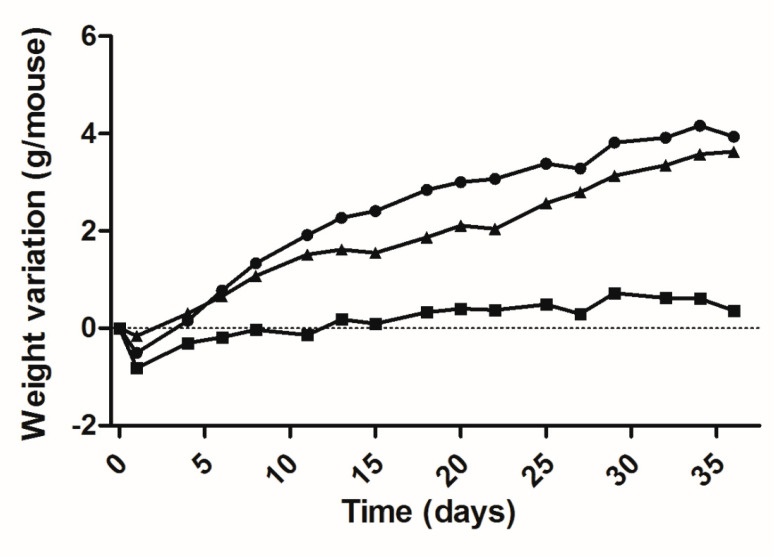
Weight variation of lit/lit mice during a 36-day bioassay, after a single administration of 50 μg mGH plasmid DNA in two sites of each tibialis cranialis muscle, followed by electrotransfer based on the 375 V/cm protocol. Equations for each curve, adjusted to the quadratic relationship Y = B_0_ + B_1_X + B_2_X^2^: (
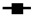
) negative control (saline), *n* = 5 mice: Y = −0.4322 + 0.0451X − 0.0004·10^−3^X^2^ (DF = 15; R^2^ = 0.7940); (
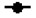
) DNA-treated mice, *n* = 7: Y = −0.3897 + 0.2256X − 0.0028·10^−3^X^2^ (DF = 15; R^2^ = 0.9808); (
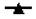
) normal (heterozygous) mice, *n* = 4: Y = −0.0535 + 0.1195X − 0.0004·10^−3^X^2^ (DF = 15; R^2^ = 0.9810).

**Figure 4 molecules-25-05034-f004:**
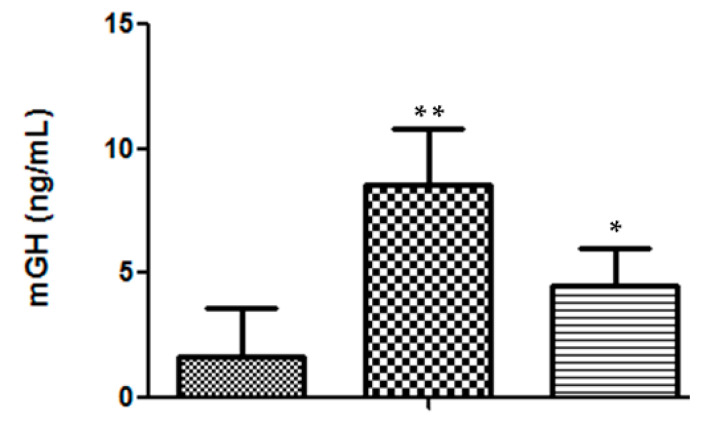
Serum mGH concentration in mice, at the end of the 36-day bioassay: (
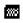
) saline, *n* = 5 mice; (
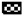
) DNA-treated mice, *n* = 7; (
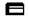
) heterozygous mice, *n* = 4. Significance test, in comparison with the negative control: * *p* < 0.05; ** *p* < 0.001.

**Table 1 molecules-25-05034-t001:** Growth parameters of lit/lit mice 36 days after administration of 50 μg of mGH plasmid DNA or saline into 2 sites of each TA muscle, followed by electroporation, in comparison with untreated heterozygous co-aged mice.

Growth Parameter	Before Treatment (mean ± SD)	After Treatment (mean ± SD)	Increase (%) * ^1^	Significance Level * ^1^	Catch-Up Growth (%)
**Body weight (g)**					
DNA-treated	10.63 ± 0.96	14.81 ± 1.25	39.3	*p* < 0.001	40.9
heterozygous	16.64 ± 1.82	20.22 ± 1.94	21.5	*p* < 0.05	
saline	10.46 ± 0.55	11.07 ± 0.76	5.8	n.s. * ^2^	
**Nose-to-tail length (cm)**					
DNA-treated	13.19 ± 0.45	14.54 ± 0.50	10.2	*p* < 0.002	38.2
heterozygous	14.55 ± 0.39	15.80 ± 0.74	8.6	*p* < 0.02	
saline	13.18 ± 0.36	13.76 ± 0.24	4.4	*p* < 0.02	
**Right femur length (mm)**					
DNA-treated	9.4 ± 0.03	12.50 ± 0.05	32.9	*p* < 0.002	82.6
heterozygous	10.8 ± 0.09	12.90 ± 0.11	19.4	*p* < 0.02	
saline	10.0 ± 0.05	10.6 ± 0.04	6.0	n.s.	
**Left femur length (mm)**					
DNA-treated	10.0 ± 0.08	12.40 ± 0.06	24.0	*p* < 0.002	76.9
heterozygous	11.1 ± 0.07	13.00 ± 0.12	17.1	*p* < 0.05	
saline	10.1 ± 0.07	10.40 ± 0.04	2.9	n.s.	

* ^1^ After treatment compared with before treatment; * ^2^ n.s., non-significant (*p* > 0.05).

**Table 2 molecules-25-05034-t002:** Serum concentrations of mIGF-1 (ng/mL) in lit/lit and heterozygous mice at the end of the 36-day bioassay.

Animal Group	n	After Treatment (mean ± SD)	Difference (%) * ^1^	Significance Level * ^1^	Catch-Up (%)
DNA-treated	7	71.2 ± 16.9	49.2	n.s. * ^2^	8.5
heterozygous	4	325.0 ± 142.0	581.3	*p* < 0.005	
saline	5	47.7 ± 33.5	-	-	

* ^1^ DNA-treated or heterozygous group compared with saline-treated lit/lit; * ^2^ n.s., non-significant (*p* > 0.05).

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
