# Peer review of "Optimization of Mouse Growth Hormone Plasmid DNA Electrotransfer into Tibialis Cranialis Muscle of “Little” Mice"

_molecules, 2020, doi:10.3390/molecules25215034_

Round 1
Reviewer 1 Report
Based on the answers of the Authors and the changes, the manuscript has been improved. It may be suitable for pubblication
Reviewer 2 Report
This is a re-submission following a first review.
The authors have addressed all the comments and suggestions I made in the first review. The quality of the article has significantly improved.
This manuscript is a resubmission of an earlier submission. The following is a list of the peer review reports and author responses from that submission.
Round 1
Reviewer 1 Report
The critical remarks in more detail:
- The discussion should be written after addressing the results. This section is often considered the most important part of a research paper because it most effectively demonstrates your ability as a researcher to think critically about an issue, to develop creative solutions to problems based on the findings, and to formulate a deeper, more profound understanding of the research problem you are studying.
- In the introduction lines 47-53, there are only references from the authors. It would be interesting to include some references from others like Catini et al. Growth Hormone & IGF Research, 2018 in order to clarify some key concepts related with the human and mouse GH genes.
- In the results, we cannot find in the Figure 1A the injected volume of the negative control. Moreover, in the Figure 1B it would be highly recommended to include the serum hGH concentration in lit/scid to compare with the mGH concentration.
- The authors describe in Figure 2 their methodology based on two injections sites in the same and different muscles. However, it is not reflected in the manuscript.
- The results obtained in the serum of lit/scid mice three days after two injections of genomic hGH DNAplasmid into each tibialis cranialis muscle are not presented, but It would be interested to include them.
- In the Figure 3, the results about Weight variation of lit/lit and heterozygous mice during a 36-day bioassay are significant?
- Line 123-124: How many days after the test are these levels maintained? You should specify it.
- Line 124-127: The authors do not refer Figure 4 in this paragraph.
- In the Table 1 and Table 2 legends do not report the parameters of heterozygous mice 36 days after administration.
Reviewer 2 Report
In this work the Authors have tried to improve the protocol for growth hormone delivery. Although the quality of the manuscript appears almost sufficient, the work is not innovative, and the results are not impressive. I suggest delaying the publication and improve the manuscript with results on younger mice, as stated in the Conclusion section.
- All figures lack indications on statistical analysis in the legends and in the graphs.
- Figure 1. Why these experiments were carried out on lit/scid mice? The work was focused on lit/lit mice. Furthermore, the number of tested animals should be increased to lower standard deviations.
- Line 125. P<0.02 significant? Is it true?
- Explain the acronyms the very first time that they appear (see Introduction and Abstract)